**Data Availability Statement:** Due to the confidentiality of the patient data, data are available upon request by qualified researchers. The data used to support the findings of this study are

# Prescribing errors and associated factors in discharge prescriptions in the emergency department: A prospective cross-sectional study

**Mona Anzan**[1,2], **Monira Alwhaibi** [2,3], **Mansour Almetwazi**[2], **Tariq M. Alhawassi**[1,2,3]*

**1** Pharmacy Services, King Saud University Medical City, Riyadh, Saudi Arabia, **2** Department of Clinical Pharmacy, College of Pharmacy, King Saud University, Riyadh, Saudi Arabia, **3** Medication Safety Research Chair, College of Pharmacy, King Saud University, Riyadh, Saudi Arabia

* Tarriq@ksu.edu.sa

## Abstract

### Objectives

Evidence regarding the prevalence of medication prescribing errors (PEs) and potential factors that increase PEs among patients treated in the emergency department (ED) are limited. This study aimed to explore the prevalence and nature of PEs in discharge prescriptions in the ED and identify potential risk factors associated with PEs.

### Methods

This was a prospective observational cross-sectional study in an ambulatory ED in a tertiary teaching hospital. Data were collected for six months using a customized reporting tool. All patients discharged from ED with a discharged prescription within the study period were enrolled in this study.

### Results

About 13.5% (n = 68) of the 504 prescriptions reviewed (for 504 patients) had at least one error. Main PEs encountered were wrong dose (23.2%), wrong frequency (20.7%), and wrong strength errors (14.6%). About 36.8% of identified PEs were related to pediatric prescriptions, followed by the acute care emergency unit (26.5%) and the triage emergency unit (20.6%). The main leading human-related causes associated with PEs were lack of knowledge (40.9%) followed by an improper selection from a computer operator list (31.8%). The leading contributing systems related factors were pre-printed medication orders (50%), lack of training (31.5%), noise level (13.0%), and frequent interruption of prescriber and distraction (11.1%). Prescribers' involved with the identified errors were resident physicians (39.4%), specialists (30.3%), and (24.4%) were made by general practitioners. Physicians rejected around 12% of the pharmacist-raised recommendations related to the identified PEs as per their clinical judgment.

restricted by the IRB (number: E-17-2551) for this
research authors, MA, MW, MM, and TA had
access to the data. For researchers interested in
working with this dataset, please contact Rubie M.
de Ocampo, E-mail: rdeocampo@ksu.edu.sa.

**Funding:** The project was fully supported
financially by the Vice Deanship of Research
Chairs, King Saud University Riyadh, Saudi Arabia.

**Competing interests:** he authors declare that there
are no conflicts of interest regarding the publication
of this paper.

## Conclusion

PEs in ED setting are common, and multiple human and systems-related factors may contribute to the development of PEs. Further training to residents and proper communication between the healthcare professionals may reduce the risk of PEs in ED.

## Introduction

A medication-related error is defined as "Failure in the treatment process that leads to, or has the potential to lead to, harm to the patient" [1]. In the United States (U.S.), it was estimated that medication errors occurred at a rate of 0.8 per 100 admissions [2]. Medication errors, in general, are associated with high morbidity and increase the length of patients' admission. Around 6 to 7% of hospital admissions globally appear to be medication related [3], which may increase healthcare costs and human loss [2, 4]. Consequently, MEs may lead to patients losing their trust and faith in the provided health care system [2, 4].

The National Coordinating Council for Medication Error Reporting and Prevention Taxonomy (NCC MERP) states that "Medication errors, may be related to professional practice, health care products, procedures, and systems, including prescribing, order communication, product labeling, packaging, and nomenclature, compounding, dispensing, distribution, administration, education, monitoring, and use" [5]. One of the most common MEs is the Prescribing Errors (PEs) which has been defined as "Medication errors initiated during the prescribing process which includes incorrect selection of medications, wrong dose (over/under therapeutic dosing), wrong strength, wrong frequency, incorrect route of administration, inadequate instruction for the use of medication and wrong dosage form" [6].

Emergency Department (ED) services in Saudi Arabia are provided by private healthcare institutes as well as the governmental healthcare institutes (including the Ministry of Health and the non-Ministry of Health institutes) that have the capability for such acute care services. The private sector has no pre-hospital ED services involvement and provides minimal services (e.g., patients transportation services for non-critical cases). The vast majority of the pre-hospital ED services, as well as the ED services, are provided by the Ministry of Health institutes and the Non- Ministry of Health institutes, although that the aforementioned sector provides ED services within specified catchment areas and represents a smaller segment of the ED services compared to the Ministry of Health institutes. The ED services are regulated by the Emergency, Disasters, and Ambulatory Transportation General Department that belongs to the Ministry of Health [7].

Medication errors (MEs) are very common at the prescribing stage, in particular in the ED setting [8]. It is estimated that at least 3% of all hospital-related adverse drug events occur in ED, due to the nature of ED, as it is one of the most commonly visited settings, which provides 24 hours' health care services [9]. A study conducted in ED at a tertiary care hospital in the U. S. showed that almost 54% of MEs occurred at the prescribing stage [10]. Another study conducted in a tertiary care hospital in India revealed that PEs occur in 16.2% of prescriptions in the ED [11]. In the Middle East, two studies conducted in a teaching hospital in Tehran revealed that 50.5% of the total MEs occurred in the ED, with 22.6% of these being PEs [12, 13]. The common types of errors were prescribing the wrong dosage, administering the drug to the wrong patient, and following physicians' oral orders [13]. A systematic review of 45 published studies from the Middle East found that 46% of the reviewed studies have reported that the most common MEs have happened at the prescribing stage of the medication use process

[14]. This review has also shown that incorrect dosing, wrong frequency, and wrong strength were the most prevalent to happen PEs [14]. A recent systematic review of 50 published studies, mainly from Iran, Saudi Arabia, Egypt, and Jordan [15] have reported that the most common contributing factors for errors among these studies were lack of knowledge, insufficient staffing levels, and heavy workload.

Although many studies evaluated the prevalence and nature of PEs in the ED settings [12, 16–19], studies in the ambulatory acute care settings, particularly in Saudi Arabia, are limited. While one study was conducted at a university teaching hospital in Riyadh, Saudi Arabia, found that MEs were common in the outpatient departments (50%), PEs accounted for 44% of the total number of reported MEs. Besides, the wrong dosing (31.3%) was found to be the most common type of MEs [20]. The high prevalence of PEs with the limited number of studies conducted in this research area has led to this study's aim to assess the prevalence and nature of all PEs in the ED and identify potential risk factors associated with increased risk of PEs.

## Methods

### Study design and setting

A prospective cross-sectional evaluation to assess PEs in emergency settings using a purposively designed data collection tool was conducted. Data collection was conducted at ambulatory ED in a large teaching hospital in Riyadh, Saudi Arabia. The hospital is approximately a 1200-bed facility with all general and subspecialty medical services. The ED is considered a level-I emergency treatment facility opened round the clock (24 hours a day, seven days a week) where a level I ED facility is known as the "center that is capable of providing total care for every aspect of injury–from prevention through rehabilitation [21].

Although the study site for this study is considered a Non- Ministry of Health institute, however similar to other governmental institutes, patients visiting the ED pass a primary station (the triage station which is covered by paramedics with general practitioners who perform the primary assessment) to patients then transfer the patient to the proper medical care when needed in the different ED units according to their age group or illness. The study site is considered a referral ED that covers cases for patients of all age groups as well as medical and surgical cases. The ED is accountable for the direct treatment of any received mild to moderate medical or surgical emergency cases in addition to life-threatening cases that may present with serious illnesses. Short-term care is provided for the received acute medical or surgical emergencies until the patient is either discharged home (a referral to the ambulatory care clinics is then given when further follow-up is required by specialists) or transferred to the inpatient setting when long term care is required.

The prescribing process in the ED is done using the hospital Electronic System for Integrated Health Information (eSiHi application). Physicians write patients' medications upon their discharge; patients then get directed to the ED pharmacy. Then the pharmacist accesses the patient's medical file for medication dispensing, checks the prescription, dispenses medication, and provides counseling to discharged ED patients.

### Study sample

The study sample consisted of patients treated and discharged by the ED. The study used the following inclusion criteria: (1) patients from all age groups and both genders, (2) patients admitted and treated in the ED, and (3) who were then discharged with an electronic prescription. Exclusion criteria were: (1) patients from other departments or wards, and (2) patients

discharged from ED without a prescription. The data collection was conducted over six months between July to December 2017.

## Sample size calculation

The sample size required for this study was calculated based on the anticipated prevalence of PEs according to previously published studies [12, 16–19] at [Z = 1.96 (5%), α = confidence level (5%), P = 0.4]. The sample size estimated for the study was 371 patients.

## Ethical approval

This study was approved by King Saud University Medical City Institutional Review Board (IRB) [approval number E-17-2551]. The IRB waived participants' consenting since this study assesses PEs and as per the good clinical practices and patients' rights conduct as any PEs have to be assessed and resolved for patient safety. Therefore, any questions that might disclose the private or personal information of patients or their identity were avoided. Treating physicians who issued the prescription during the data collection period were verbally informed about the study. Moreover, to protect the patient data's confidentiality, only authors of the research, MA, MW, MM, and TA had access to the data. The eSiHi application accessibility is highly restricted only to authorized staff in the study site.

## Questionnaire development and data collection

The data collection form was developed using the National Coordinating Council for Medication Error Reporting and Prevention (NCC MERP) Taxonomy of Medication Errors matching the objectives of the study [22]. The data collection tool was then piloted by the researcher MA on a sample of 20 patients and modified accordingly based on the pilot phase results to reach the final form by the research authors (Appendix I). Questions were then combined and assessed as a group until consensus was reached to finalize the study data collection tool. The data collection tool was then piloted by the researcher MA on a sample of 20 patients and then modified accordingly based on the results of the pilot phase to reach the final form by the research authors. The study's final data collection tool was then redesigned as an electronic data collection form. The study research pharmacist who was covering the ED pharmacy during the evening shift worked on viewing and evaluating all patients' prescriptions received. If the prescription was identified as a potential PEs, further evaluation using the data collection form was done to explore the type of error and associated information. Identified potential PEs were then discussed directly with the prescriber using either verbal or written communication tools including the pharmacist recommend the intervention to correct the identified potential PEs. Prescriber's response to the pharmacist recommended intervention was recorded as either an accepted or rejected intervention. In the rejected intervention, the physicians provide a rationale and supporting evidence (using the institute official forms that the physician has to fill and attach with supporting evidence) (Fig 1). All interventions or recommendations related to the identified potential PEs were documented and submitted to the institute's Medication Safety Officer for validation and quality assessment. This process is done to assure safe medical practice and therefore patient safety according to the institute policy and procedures and proper actions that act hard to improve the "learning Culture" environment.

The data collection included patient demographic data (such as age, gender, weight, etc.) and data related to the identified PEs. Data about the identified PEs including encountered setting, error type, medications' information involved with the PE (such as dosage form, therapeutic class, strength, description of the error, etc.), healthcare professionals involved in

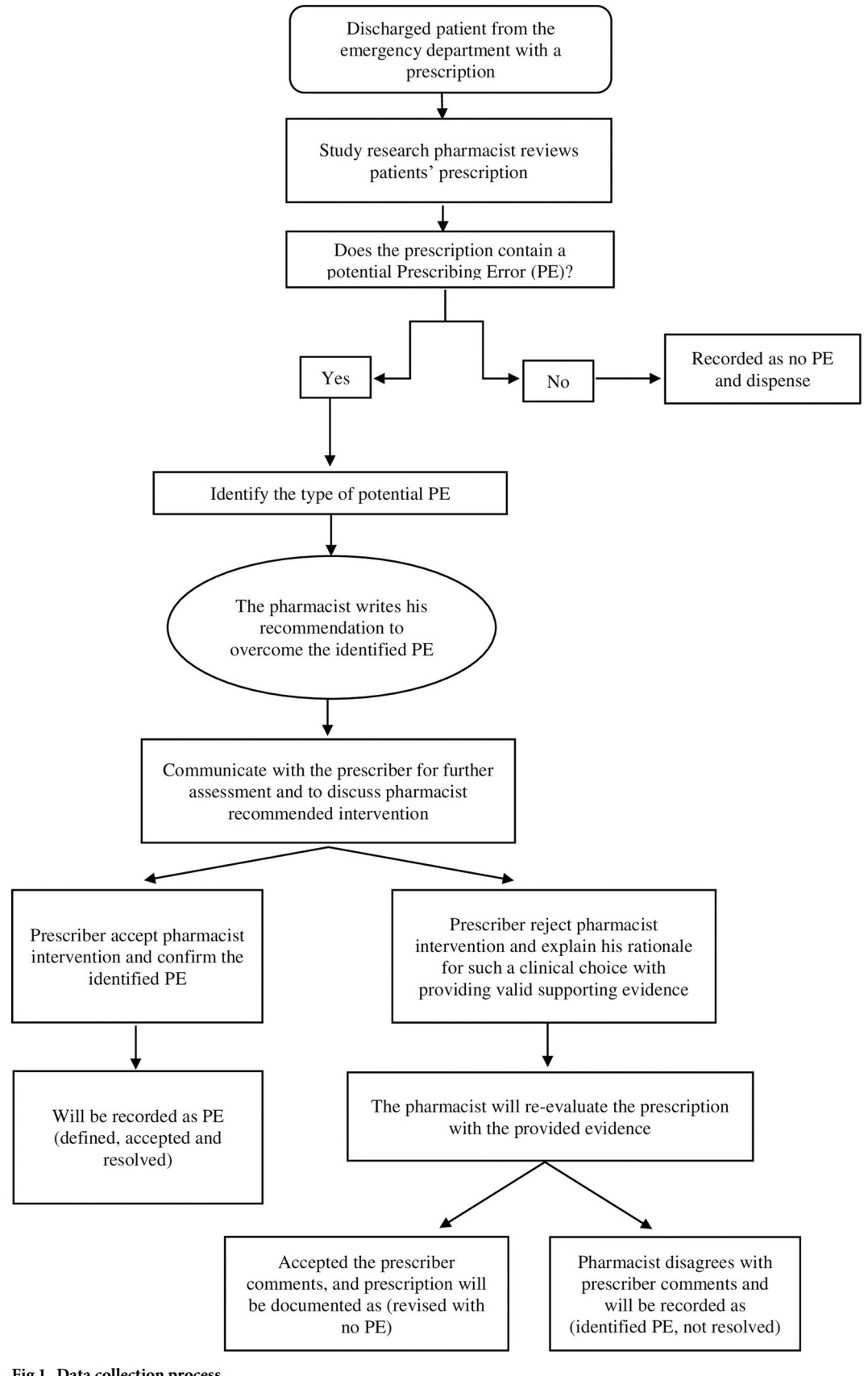

**Fig 1. Data collection process.**

identified PEs, and other data related to the potential causes and contributing factors using the study modified NCC MERP Taxonomy of Medication Errors tool.

## Statistical analysis

Collected data were coded and entered for analysis using the Statistical Package for Social Science Students (SPSS 20.0) (IBM Corp., Armonk, N.Y., USA). Descriptive statistics were used to illustrate demographical characteristics. Categorical variables were presented as frequencies and percentages.

## Results

A total of 504 prescriptions were observed and assessed during the study period for 504 patients (one prescription/patient), and 68 (13.5%) of patient' prescriptions were identified with confirmed PEs. The total number of PEs was 82; where some prescriptions have more than one PEs per patient prescriptions (e.g., prescriptions had both the wrong dose and wrong route of administration). For patients with identified PEs, an equal ratio of females (50.0%) to males was found. About 63.2% of the identified PEs were among adult patient prescriptions, while 36.8% were in pediatric prescriptions. Among adult patient prescriptions, 26.5% of the identified PEs was from the acute care emergency unit, 20.6% were from the initial emergency management unit (Triage), and 7.4% occurred in the flu clinic while obstetrics and gynecology emergency had a lesser percentage of PEs with a percentage of 5.4%. It has to be noted that the wrong strength/concentration, wrong dosage form, and wrong route of administration were higher in the adults' population compared to the pediatrics.

### Type of prescribing errors

Prescribing errors encountered were categorized according to their types. The most commonly observed PEs were wrong dose (e.g., four years old patient weighed 13 kg, prescribed Tylenol III [Paracetamol 500mg–codeine 30mg] as 1 tablet) followed by wrong frequency (e.g., Amoxicillin / Clavulanic acid 1g was prescribed as 1 tablet 16 times per day) (Table 1). The most

**Table 1. Identified prescribing errors as per type.**

| Type of Prescribing Error | Total Sample | |
|---|---|---|
| | N | % |
| **Total Number of PEs**[*] | 82 | 100.0 |
| Wrong dose | 19 | 23.2 |
| Wrong Frequency | 17 | 20.7 |
| Wrong Strength/Concentration | 12 | 14.6 |
| Wrong Dosage Form | 8 | 9.8 |
| Wrong patient | 6 | 7.3 |
| Dose omission (if a medication has not been prescribed) | 5 | 6.1 |
| Wrong Drug | 3 | 3.7 |
| Wrong Duration | 3 | 3.7 |
| Wrong Route of Administration | 3 | 3.7 |
| Documented allergy | 2 | 2.4 |
| Drug-Drug interaction | 2 | 2.4 |
| Drug-Disease interaction | 1 | 1.2 |
| The drug is not indicated (the drug does not treat the diagnosis or not indicated for such use) | 1 | 1.2 |

**ote:** Some Prescriptions have more than one prescribing error
**PEs:** Prescribing Errors

encountered therapeutic class of medication associated with PEs was analgesic medications (33.8%) followed by antibiotics (29.0%), gastrointestinal medications (9.6%), and allergy relief medications (6.4%) (Table 2). Most of the identified PEs occurred with the tablet dosage form, followed by oral liquids (Table 2).

## PEs causes and contributing factors

This study found that the primary sources of PEs were "human-related" causes, followed by "system-related" contributing factors (Table 3). The leading human-related causes for PEs were lack of knowledge followed by an improper selection from a list by the computer operator, and insufficient training to use the electronic system correctly (Table 3). The leading contributing systems related factors were pre-printed medication orders, lack of training on using the electronic system, the noise level, and the frequent interruption of prescriber and distraction (Table 3). Prescribers involved with the identified PEs were 39.4% residents, 30.3% specialists, 24.3% general practitioners, 4.5% nurses, and 1.5% by others (Table 4). In this study, 88.0% of the identified PEs in this study were resolved by the pharmacist and were recorded as accepted interventions, while physicians have rejected 12.0% of the raised recommendations as per their clinical judgment. In these 12% rejected interventions, pharmacist had re-evaluated the prescription to categorize these into [1. Revised with no PE] [2. identified PE, not resolved]. Of the rejected interventions, 37.0% were related to PEs in pediatric prescriptions, 26.5% were related to acute care emergency prescriptions, and 20.6% were rejected recommendations by the triage emergency unit.

**Table 2. Most common therapeutic agents classified as therapeutic classification and dosage forms involved with identified prescribing errors.**

| Therapeutic classifications | N | % |
|---|---|---|
| Analgesic, antipyretic, anti-inflammatory (painkillers) | 21 | 33.9 |
| Antibiotic | 18 | 29.0 |
| GI agents (laxatives, antidiarrheal, antispasmodics) | 6 | 9.7 |
| Allergy relief medications (systematic & topical) | 4 | 6.5 |
| Nasal decongestant (systematic & topical) | 4 | 6.5 |
| Antacid–Protein Pump Inhibitors | 3 | 4.8 |
| Antifibrinolytic | 1 | 1.6 |
| Antihistamine | 1 | 1.6 |
| Antihypertensive | 1 | 1.6 |
| Antiviral | 1 | 1.6 |
| Bronchodilator & respiratory agents | 1 | 1.6 |
| Hormonal replacement therapy | 1 | 1.6 |
| **Dosage Forms** | **N** | **%** |
| Oral formulations (Tablet, Extended-release tablet, Capsule) | 37 | 59.68 |
| Oral Liquid | 10 | 16.13 |
| Eye Drops | 6 | 9.68 |
| Cream-Ointment-Gel-Paste | 4 | 6.45 |
| Rectal | 2 | 3.23 |
| Aerosol (spray and metered) | 1 | 1.61 |
| Injectable | 1 | 1.61 |
| Others | 1 | 1.61 |

**Note:** The numbers may not add up to the total number of prescribing errors, as the table not included all types of Pes

**Table 3. Factors associated with increased risk of prescribing errors.**

| Systems Related Contributing factors | Number | % |
|---|---|---|
| Pre-printed medication orders | 27 | 42.9 |
| Training | 17 | 27.0 |
| Noise Level | 7 | 11.1 |
| Frequent Interruptions and distractions | 6 | 9.5 |
| Lack of availability of health care professional | 2 | 3.2 |
| Other | 2 | 3.2 |
| Communication systems between health care practitioners | 1 | 1.6 |
| **Human Related Contributing factors** | **Number** | **%** |
| Knowledge Deficit | 27 | 32.5 |
| Incorrect selection from a list by the computer operator | 21 | 25.3 |
| Undertrained to use the electronic system correctly | 9 | 10.8 |
| Human factors | 4 | 4.8 |
| Miscalculation of Dosage | 4 | 4.8 |
| Misinterpretation of the order | 4 | 4.8 |
| Stress (high volume workload, etc.) | 4 | 4.8 |
| Name Confusion | 3 | 3.6 |
| Transcription Error | 3 | 3.6 |
| Written/electronic miscommunication | 3 | 3.6 |
| Computer Error | 1 | 1.2 |

**Note:** Some Prescriptions have more than one factor that lead to the PE

## Discussion

This study found that 13.5% of patient's electronic prescriptions in ED had at least one error PEs. This finding is considered compatible with the low range of PEs found by other published studies that assessed PEs in the ED setting where PEs were found to occur in 13.4 to 50.5% of prescriptions in the ED [11, 23]. For instance, a recent study that evaluated PEs in a sample of 1000 prescriptions PEs in the ED in an academic medical center in the U.S. in 2017 found similar results to our study's finding with 13.4% of prescriptions in the ED had at least one PEs [23]. This percentage tends towards the lower range reported by several studies, but PEs are nonetheless alarmingly common in the ED.

In our study, 36.8% of prescriptions in the pediatric setting had PEs where wrong dose, wrong frequency, wrong concentrations, and wrong administration route were the most commonly identified types of PEs, which form a challenge for practitioners and patients. The pediatric emergency is recognized as a high-risk environment for MEs due to multiple factors such as the lack of standardized pediatric drug dosing and formulations, weight-based dosing, and the numerous common transitions of care [24].

**Table 4. Healthcare professionals involved in prescribing errors.**

| Job Title | N | % |
|---|---|---|
| General Practitioner | 16 | 24.30 |
| Specialist | 20 | 30.30 |
| Resident | 26 | 39.40 |
| Nurse | 3 | 4.50 |
| Others | 1 | 1.50 |

The prescribed therapeutic agents' wrong dosage was considered a predominant error with a high percentage in our study with Antibiotics dosing. This was also found high by other studies; for instance, among the pediatric the of PEs was (22.7%) compared to adults (11.7%) where PEs with antibiotics found the most common [23]. Another study done in the U.S. over one year on 18 pediatric EDs has also shown that the most commonly reported ME were anti-infective agents and analgesics, followed by other medications such as intravenous fluids and respiratory medications [25]. Therefore, clinical aid solutions to assist prescribers with medication prescription (e.g., dose calculators and clinical pharmacists involvement in ED for pediatric patients in the ED) need to be investigated and considered [26].

This study found that the highest number of PEs was done by ED residents similar to the published findings of other studies [23]. Having a large percentage of PEs generated from residents highlights the need for well-designed educational as well as antimicrobial stewardship programs to improve their prescribing and reduce the risk of PEs. A published study of residents PEs in different specialties from a pediatric clinic has found that training programs which involve pharmacist and initiatives to prevent medication errors was associated with a lower rate of PEs in some specialties [27]. Pharmacists can play an essential role in reviewing prescriptions retrospectively to pinpoint the exact area of knowledge defects and more probably to provide individual feedback to those residents in the various specialties to lower PEs with more training on how to overcome PEs risk.

There are several human and systems-related contributing causes and factors to PEs in the emergency setting. Common human-related causes for PEs were lack of knowledge followed by an improper selection from a list by the computer operator. This was also found in the literature where the prescriber's limited knowledge about the medication prescribed was more associated with PEs risk for patients being treated in the ED [28]. The primary contributing systems related factors were pre-printed medication orders and the lack of training. A study published by the BMJ journal assessing the impact of preprinted prescription forms on medication PEs found that pre-printed prescription form has the potential to decrease certain medications related PEs [29]. However, new error types can occur, which is maybe the case in our study. In fact, computerized physician ordering entering (CPOE) systems have not entirely eliminated medication errors. For instance, CPOE systems may fail to address critical dosing requirements due to providers' tendency to override prescribing alerts [24]. This kind of defect does not neglect the importance of CPOE as it can prevent millions of medication errors from happening if used efficiently compared to the old fashion handwritten prescription [24]. Further research is needed to rigorously explore the problem and assess the development of inter-graded prescribing aids that considers both; prescribers and the use of CPOE systems to prevent or minimize PEs in such a setting. On the other hand, there is also an increasing need to fully consider other potential contributing factors that may lead to PEs including prescribers' knowledge, training on CPOE use, and the availability of a right practicing environment to reach better patient safety outcomes by reducing medication related errors.

Multiple practical and research implications can emanate from the present study findings. The involvement of pharmacists in the medication use process, especially in ED, could significantly influence reducing the rate of PEs Incidents. Studies have reported that a pharmacist in the ED could help reduce medication errors by optimizing pharmacotherapy, improving patient safety, educating patients and clinicians [28, 30, 31]. A cohort study has shown that with the absence of pharmacists, over 13 folds more errors encountered in the ED than with pharmacists present. Another study evaluating MEs rate before and after pharmacist involvement in an ED had concluded that the percentage of MEs was decreased by two-third in the intervention group (with pharmacists has been involved) compared with the control group (no pharmacist was involved) [28]. Also, another study of pharmacist intervention in the ED

in Spain found that drug therapy services provided by clinical pharmacists were significantly correlated with identifying PEs including the serious errors in the ED [30].

The study findings emphasized the importance of conducting future research to evaluate the negative clinical, economic and humanistic impacts that PEs can cause in the ED which is always described as a fast working environment that makes it a high-risk area for medication errors including PEs. Furthermore, research to address the positive impact of preventing PEs; cost saving impact and role of pharmacists for the ED patient and the impact of prevented errors on patients' quality of life or preventing new health problems in patients' lives is needed. Yet, this study may add to the current data and work as a base for future research exploring PEs in the emergency setting. Further, this study had tried to pinpoint potential contributing factors that may increase the risk of PEs in the ED and therefore shall help future research to consider these factors and what solutions are needed.

This study has some limitations. It was conducted in a single ED in a tertiary hospital therefore; the findings cannot be generalized to all ED in Saudi Arabia. The study was conducted in a hospital that uses an electronic prescribing system. Therefore, comparisons with other hospitals where a handwritten prescription system is used may provide different findings and more insightful views on PEs. Another limitation that data were collected in the evening shift only; thus, this might limit the study findings' generalizability although even shift is considered a busy shift and the study site is a large ED setting and patients are seen in large numbers around the clock during the whole weekdays. In addition to the previously listed limitations, the study design is a descriptive study and no data was collected about health outcomes of PEs in the ED. Besides, no information was collected about the chronic illnesses of the patients involved in this study. Also, the categorization of medication errors according to the severity of the patient's health conditions was not taken into account in this study. Further, the study was limited by the lack of a denominator for the data provided. For example, a comparison of error rates for each medication would provide a complete picture of what types of prescriptions were more likely to have errors.

## Conclusions

PEs in the emergency setting are common where multiple human and system-related factors may increase the risk of the development of PEs. This study highlights the need for future research to explore the role of clinical aid supporting tools in addition to prescribing education and training programs to improve prescribing knowledge and skills particularly for junior doctors. While pharmacists' involvement in ED can play a vital role in reviewing prescriptions and therefore help reduce PEs, strategies to mitigate the impact of other system-related risk factors are warranted.

## Supporting information

**S1 Appendix. Data collection tool.**
(DOCX)

## Author Contributions

**Conceptualization:** Mona Anzan, Monira Alwhaibi, Mansour Almetwazi, Tariq M. Alhawassi.

**Data curation:** Mona Anzan, Monira Alwhaibi, Mansour Almetwazi, Tariq M. Alhawassi.

**Formal analysis:** Mona Anzan, Monira Alwhaibi, Tariq M. Alhawassi.

**Funding acquisition:** Mona Anzan.

**Investigation:** Mona Anzan, Monira Alwhaibi, Mansour Almetwazi.

**Methodology:** Mona Anzan, Monira Alwhaibi, Mansour Almetwazi.

**Project administration:** Mona Anzan, Tariq M. Alhawassi.

**Writing – original draft:** Mona Anzan, Monira Alwhaibi, Mansour Almetwazi, Tariq M. Alhawassi.

**Writing – review & editing:** Mona Anzan, Monira Alwhaibi, Mansour Almetwazi, Tariq M. Alhawassi.

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
