## [Decision Letter · Decision Letter 0]

15 Apr 2020

PONE-D-20-06984

Prescriptions Errors Prevalence and Associated Factors in an Emergency Setting: A Prospective Cross-Sectional Study

PLOS ONE

Dear Dr Alhawassi,

Thank you for submitting your manuscript to PLOS ONE. After careful consideration, we feel that it has merit but does not fully meet PLOS ONE’s publication criteria as it currently stands. Therefore, we invite you to submit a revised version of the manuscript that addresses the points raised during the review process.

This paper describes a very interesting study. However, as the reviewers specify, it presents several major problems. Please, revise your manuscript and address all their valuable comments.

In addition to the remarks of the reviewers, I observe other points that need to be clarified:

- I am not familiar with the health care organization in Saudi Arabia. A description of the different clinics or units that compose the Emergency Department will be helpful.

- The enrolment criteria for patients and the dates of the observation period should be reported

- I have found some inconsistencies in the results. According to the authors, they identified 68 prescription errors. In table 1, the total number of “type of prescribing errors” is 83 (we can assume that a prescription could contain more than one error). However, in table 2 the total amount of “Involved Prescriptions” by “Therapeutic classification” are only 62. The same number (62) is the sum of “Dosage forms” in table 2 and the “Contributing factors” in table 3.

On the other hand, the authors should observe the Criteria for Publication in PLOS ONE. Remember that PLOS journals require authors to make all data necessary to replicate their study’s findings publicly available without restriction (https://journals.plos.org/plosone/s/data-availability). Besides, the manuscripts should conform the corresponding reporting guidelines. Revise the information on the web site (https://journals.plos.org/plosmedicine/article?id=10.1371) and complete the STROBE checklist for observational studies.

We would appreciate receiving your revised manuscript by May 30 2020 11:59PM. To enhance the reproducibility of your results, we recommend that if applicable you deposit your laboratory protocols in protocols.io, where a protocol can be assigned its own identifier (DOI) such that it can be cited independently in the future. For instructions see: http://journals.plos.org/plosone/s/submission-guidelines#loc-laboratory-protocols

We look forward to receiving your revised manuscript.

Kind regards,

Juan F. Orueta, MD, PhD

Academic Editor

PLOS ONE

Journal Requirements:

1. Please provide additional details regarding participant consent. In the Methods section, please ensure that you have specified (1) whether consent was informed and (2) what type you obtained (for instance, written or verbal). If your study included minors, state whether you obtained consent from parents or guardians. If the need for consent was waived by the ethics committee, please include this information.

2. The Data collection tool provided in the Appendix has previously been copyrighted. Please ensure this is removed - a reference to the tool is sufficient.

Reviewers' comments:

Reviewer's Responses to Questions

**Comments to the Author**

1. Is the manuscript technically sound, and do the data support the conclusions?

Reviewer #1: Partly

Reviewer #2: Yes

2. Has the statistical analysis been performed appropriately and rigorously? 

Reviewer #1: No

Reviewer #2: Yes

3. Have the authors made all data underlying the findings in their manuscript fully available?

Reviewer #1: Yes

Reviewer #2: Yes

4. Is the manuscript presented in an intelligible fashion and written in standard English?

Reviewer #1: No

Reviewer #2: Yes

5. Review Comments to the Author

Reviewer #1: The study highlights an important topic not previously addressed in Saudi Arabia. However the paper needs extensive revision/ proof reading for grammatical errors, punctuation and sentence structuring. An example is the first paragraph in the introduction.

Introduction:

Line 100- do you mean This study…..

Methods:

Would be useful to provide the readers more background information about the settings in relation to the prescription pathway ( electronic prescribing, goes through pharmacy for checks, patient collects medication from pharmacy etc).

Line 121: please cite and reference ‘the previously published studies’

Ethical approval- line 123- more information needs to be provided regarding obtaining patient consent, patient anonymity and confidentiality?

Line 134: was the pharmacist collecting the data involving in the pilot stage as well in any way?

Lines 139-140: sentence is very long and unclear- needs to be clarified.

Line 140: you mention including patient’s file number and initials. How was anonymity of the patients protected? This needs to be addressed in the ethical approval section-

Line 145: Why was the data only collected in the evening? Does this mean any patients discharged during the day were excluded from the study? This is unclear.

Line 150- what is the study questioner?

Line 156-157: The study is a prospective study- So my understanding is that any prescribing error was identified and resolved before the medication was dispensed to the patient? how were the recommendation rejected by assuring that the PEs was corrected, and no harm has reached the patient? Unclear- I assume this is for previous known patients or repeat prescriptions rather than new patients? Or do you mean that the patient was receiving the medication and was OK while in the ED and thus discharged with the same medication/ dose? Needs clarification.

Line 164: continuous?

Unclear how the causes of PE were identified? How did the pharmacist/researcher identify what were the factors contributing to the errors? Did the pharmacist ask the prescriber for the reason for making an error?

How was any disagreement dealt with if the PE was rejected by the prescriber?

How was the validity and reliability of the data collected assured?

Results:

So how many patients were recruited for the study (were 371 patients recruited?) You only mention the number of prescriptions and not patients. Analysis of the data could further be presented for the total number of patients besides the number of prescriptions.

Line 196: You mention that 12% of the interventions were rejected. Where they justifiable?

Good discussion. Would be interesting to report if these potential or near miss errors were/are reported? You talk about educating the prescribers but no mention whether incident reporting is encouraged or not and the role it can play to reduce errors and promote safety culture.

Reviewer #2: The article has scientific rigor and importance to the world literature and Saudi Arabia. However, for the manuscript to be accepted in this Journal, I strongly advise that changes must be made throughout the text and, if possible, new data and discussions must be added. Thus, these some changes will make the article more robust and understandable to readers.

Below, I list the points that need to be improved.

Title

- Change the term "Emergency Setting" to "Emergency Department".

Introduction

- The Introduction is short and focused.

- Currently, there is a confusion in the understanding of the terms "Medication errors" and "adverse drug events". In this article, the authors do not study "adverse drug events", so I suggest deleting this information.

- The first paragraph (lines 69-73) provides a situational panorama with very old and obsolete studies. In general, articles older than 5 years should not be used. Please include new and updated references, so that the reader can understand the real context of medication errors.

- Clarify the phrase "high morbidity and increase the length of patients' admission and hospital stay" (line 71-72). Specify how much "increase" and “high” there was.

- The phrase "Limited studies have focused on the prevalence and nature of PEs in the ED" (lines 93-94) is inaccurate information. I suggest the authors do an extensive search in the literature. There are many articles published about the subject of the article.

- To improve the justification of the article, I suggest that you address the importance and consequences (negative impacts) that prescription errors can cause in an emergency department.

Methods

- The methods section is missing key information about the study.

- Did the authors collect sociodemographic information, illnesses or complaints from the patient or why he / she sought the emergency department?

- a fundamental question: Have data been collected on the positive impact of preventing errors? Did the prevented errors save costs for the Emergency Department or for the patient? Did the prevented errors improve the patients' quality of life? Did mistakes prevent new health problems in patients' lives?

- Clarify who the research team is.

Results

- It would be very important for the authors to present the clinical, eco-economic and humanistic impacts of preventing errors.

- Table 3: change “contributing factors” to “contributing factors (systems related)” and “causes for identified prescription errors” to “causes for identified prescription errors (human-related)”.

Discussion

- Overall, the discussion section requires clear connection with the study findings. Further tightening the connection between results with discussion will provide clear understanding of the study contribution and future direction of prevention of medication errors in Emergency Department.

- It would be very important to relate the complaint or reason for going to the emergency department with the use of medicines.

- If clinical, economic and humanistic impacts have not been collected, I suggest adding it as a study limitation.

- The phrase “Although this study is one of few studies that evaluated the PEs in the ED” (line 265) is inaccurate information. There are several studies in the literature.

Conclusion

- The conclusion is very short. Explore further the conclusions that you observed with the results and describe the implications that this can have.

References

- Only 5 (22,7%) studies were published in the last 5 articles. This is a problem.

6. PLOS authors have the option to publish the peer review history of their article (what does this mean?). If published, this will include your full peer review and any attached files.

Reviewer #1: No

Reviewer #2: Yes: Genival Araujo dos Santos Júnior

---

## [Author Response · Author response to Decision Letter 0]

11 May 2020

We would like to thank the editor and the reviewers for the time and effort they spent on reviewing our manuscript entitled “Prescriptions Errors Prevalence and Associated Factors in an Emergency Department: A Prospective Cross-Sectional Study", their valuable and insightful comments have improved our manuscript substantially.

We are excited to have been given the opportunity to revise our manuscript and respond to the revisions. We have gone through all comments received and appropriate changes/amendments have been made correspondingly to the paper (Highlighted) are summarized in the following:

Editor Comments to Author

In addition to the remarks of the reviewers, I observe other points that need to be clarified 

Comment # 1: I am not familiar with the health care organization in Saudi Arabia. A description of the different clinics or units that compose the Emergency Department will be helpful.

Response: Thank for this valuable suggestion, we have added more description about the ED services in Saudi Arabia in the background (Pages 3-4, lines 82-92) to inform the readers about this point and in the methods section a paragraph was added explaining the study site ED (Page 5, lines 120-135).

Comment # 2: The enrolment criteria for patients and the dates of the observation period should be reported.

Response: The study enrollment criteria, as well as exclusion criteria, were added (Page 6, lines 144-149). 

Comment # 3: I have found some inconsistencies in the results. According to the authors, they identified 68 prescription errors. In table 1, the total number of “type of prescribing errors” is 83 (we can assume that a prescription could contain more than one error). However, in table 2 the total amount of “Involved Prescriptions” by “Therapeutic classification” are only 62. The same number (62) is the sum of “Dosage forms” in table 2 and the “Contributing factors” in table 3.

Response: Thank you and for pointing out this very important point. In this study, a total number of 504 prescriptions (this equal to 504 patients since every patient visits the ED comes with one electronic prescription) were screened and we have identified 68 confirmed prescriptions errors (68 patients). Some of these prescriptions were identified with multiple prescribing errors therefore the total number of identified prescribing errors was 82 as listed in table 1. Table 2 lists the most commonly involved medications and dosage forms with the identified errors if the error was associated with the medication itself. Table 3 lists potential factors involved with developing prescription errors either as system-related factors or human-related factors. Table 3 should not be summed as some of the identified prescription errors were linked to more than one factor making the total does not equal 100. This is now clarified in the text under results and all tables were edited for language to make the interpretation easier for readers. We highly appreciate your comment on this part of the manuscript.

Comment # 4: Please provide additional details regarding participant consent. In the Methods section, please ensure that you have specified (1) whether consent was informed and (2) what type you obtained (for instance, written or verbal). If your study included minors, state whether you obtained consent from parents or guardians. If the need for consent was waived by the ethics committee, please include this information.

Response: Thanks for this valuable point. A paragraph answering this point was added appropriately to the manuscript (Page 7, lines 156-164).

Comment # 5: The Data collection tool provided in the Appendix has previously been copyrighted. Please ensure this is removed - a reference to the tool is sufficient.

Response: Thank you for pointing this out. We have removed the data collection tool from the Appendix.

Comment # 6: We note that you have indicated that data from this study are available upon request. PLOS only allows data to be available upon request if there are legal or ethical restrictions on sharing data publicly. For information on unacceptable data access restrictions, please see http://journals.plos.org/plosone/s/data-availability#loc-unacceptable-data-access-restrictions.

Response: The data used to support the findings of this study are restricted by the study site IRB (IRB approval number: E-17-2551) to protect the confidentiality of the patient data. Only the authors of the research, MA, MW, MM, and TA had access to the data.

Reviewer(s)' Comments to Author

Reviewer# 1

The study highlights an important topic not previously addressed in Saudi Arabia. However the paper needs extensive revision/ proof reading for grammatical errors, punctuation and sentence structuring. An example is the first paragraph in the introduction.

Comment # 1: Introduction:

Line 100- do you mean This study…..

Response: Thank you for noticing this typo, we have now corrected it appropriately (Page 4, line 112). Moreover, the manuscript was entirely reviewed for English language and grammatical errors appropriately. 

*Methods

Comment # 2: Would be useful to provide the readers more background information about the settings in relation to the prescription pathway (electronic prescribing, goes through pharmacy for checks, patient collects medication from pharmacy etc).

Response: Thank you for pointing this out, now we have added more description about the settings in relation to the prescription pathway (Page 6, lines 138-142).

Comment # 3: Line 121: please cite and reference ‘the previously published studies’

Response: Thank you for this comment; we have added the required references (Page 6, line 153).

Comment # 4: Ethical approval- line 123- more information needs to be provided regarding obtaining patient consent, patient anonymity and confidentiality?

Response: Thank for the important suggestion, we have added more information under the Ethics Approval section (Page 7, lines 156-164).

Comment # 5: Line 134: was the pharmacist collecting the data involving in the pilot stage as well in any way?

Response: Thank you for raising this important point. Yes, the research pharmacist [MA] who is considered a senior registered and practicing ED pharmacist performed the data collection and was the one also involved in the pilot stage for data collection consistency. This now clarified under the “Questionnaire Development and Data Collection” section (Page 7, lines 171-174). 

Comment # 6: Lines 139-140: sentence is very long and unclear- needs to be clarified.

Response: Appropriate changes have been made to the mentioned sentence to make the sentence clearer for the readers. 

Comment # 7: Line 140: you mention including patient’s file number and initials. How was anonymity of the patients protected? This needs to be addressed in the ethical approval section-

Response: Now, we have included a statement about the confidentiality of patients’ data (Page 7, lines 161-164). 

Comment # 8: Line 150- what is the study questioner?

Response: This was a typo mistake; we mean the “study data collection tool”. This is now corrected appropriately. 

Comment #9: Line 156-157: The study is a prospective study- So my understanding is that any prescribing error was identified and resolved before the medication was dispensed to the patient? how were the recommendation rejected by assuring that the PEs was corrected, and no harm has reached the patient? Unclear- I assume this is for previous known patients or repeat prescriptions rather than new patients? Or do you mean that the patient was receiving the medication and was OK while in the ED and thus discharged with the same medication/ dose? Needs clarification.

Response: Thank you for your highly important comment. An appropriate clarification was added (Page 8, lines 177-192) using also Figure 1 which we hope this will now make reading the data collection part related to the identification of PEs point clearer for the reader. 

Comment # 11: Line 164: continuous?

Unclear how the causes of PE were identified? How did the pharmacist/researcher identify what were the factors contributing to the errors? Did the pharmacist ask the prescriber for the reason for making an error?

How was any disagreement dealt with if the PE was rejected by the prescriber?

How was the validity and reliability of the data collected assured?

Response: Thank for pointing this out. As mentioned in our response to the previous comment and to answer these comments and make the PEs identification process used in this study more clearly described, more description to the methods section was added (Page 8, lines 177-192).

*Results

Comment # 12: So how many patients were recruited for the study (were 371 patients recruited?) You only mention the number of prescriptions and not patients. Analysis of the data could further be presented for the total number of patients besides the number of prescriptions.

Response: Thank you for this comment. 371 patients were the required sample based on our study sample size calculation, however, a total of 504 patients (504 patient prescriptions as a term is used alternatively in the manuscript which indicates also the number of patients since that each patient comes to the ED pharmacy with one electronic prescription) were included (Page 8, line 184). 

Comment # 12: Line 196: You mention that 12% of the interventions were rejected. Where they justifiable?

Response: Yes, with each rejected intervention the prescriber was asked to explain to the pharmacist his supporting rationale and evidence for such a clinical choice with providing valid supporting evidence. Now we have added this description to the manuscript (Page 8, lines 177-192).

*Discussion

Comment # 13: Good discussion. Would be interesting to report if these potential or near miss errors were/are reported? You talk about educating the prescribers but no mention whether incident reporting is encouraged or not and the role it can play to reduce errors and promote safety culture.

Response: Thank you and your comment is highly appreciated. The added paragraph (page 8, lines 177-192) should answer this important point reporting the identified prescribing errors. 

Reviewer# 2

The article has scientific rigor and importance to the world literature and Saudi Arabia. However, for the manuscript to be accepted in this Journal, I strongly advise that changes must be made throughout the text and, if possible, new data and discussions must be added. Thus, these some changes will make the article more robust and understandable to readers.

Below, I list the points that need to be improved.

*Title

Comment # 1: Change the term "Emergency Setting" to "Emergency Department".

Response: Thank you and change has been made. 

*Introduction

Comment # 2: The Introduction is short and focused.

- Currently, there is a confusion in the understanding of the terms "Medication errors" and "adverse drug events". In this article, the authors do not study "adverse drug events", so I suggest deleting this information.

Response: Thank you for pointing this out, we have deleted this information from the introduction.

Comment # 3: The first paragraph (lines 69-73) provides a situational panorama with very old and obsolete studies. In general, articles older than 5 years should not be used. Please include new and updated references, so that the reader can understand the real context of medication errors.

Response: We agree with the reviewer. Now, we have updated the references and added these new references:

2.Choi I, Lee S-M, Flynn L, Kim C-m, Lee S, Kim N-K, et al. Incidence and treatment costs attributable to medication errors in hospitalized patients. Research in Social and Administrative Pharmacy. 2016;12(3):428-37.

4.Wittich CM, Burkle CM, Lanier WL. Medication Errors: An Overview for Clinicians. Mayo Clinic Proceedings. 2014;89(8):1116-25.

Comment # 4: Clarify the phrase "high morbidity and increase the length of patients' admission and hospital stay" (line 71-72). Specify how much "increase" and “high” there was.

Response: Thank you for this comment and now we have added the numbers appropriately (Page 3, line 68).

Reference:

World Health Organization. Medication errors: World Health Organization; 2016.

Comment # 5: The phrase "Limited studies have focused on the prevalence and nature of PEs in the ED" (lines 93-94) is inaccurate information. I suggest the authors do an extensive search in the literature. There are many articles published about the subject of the article.

Response: Thank you for pointing this out, we have modified this statement and added the references after doing an extensive literature search (Page 4, lines 106-107).

References:

13. Murray KA, Belanger A, Devine LT, Lane A, Condren ME, editors. Emergency department discharge prescription errors in an academic medical center. Baylor University Medical Center Proceedings; 2017: Taylor & Francis.

14. Dabaghzadeh F, Rashidian A, Torkamandi H, Alahyari S, Hanafi S, Farsaei S, et al. Medication errors in an emergency department in a large teaching hospital in Tehran. Iranian journal of pharmaceutical research: IJPR. 2013;12(4):937.

15. Alanazi MQ, Al-Jeraisy MI, Salam M. Prevalence and predictors of antibiotic prescription errors in an emergency department, Central Saudi Arabia. Drug, healthcare and patient safety. 2015;7:103.

16. Flynn EA, Barker K, Barker B. Medication-administration errors in an emergency department. American Journal of Health-System Pharmacy. 2010;67(5):347-8. 

17. Shitu Z, Aung MMT, Kamauzaman THT. Prevalence and characteristics of medication errors at an emergency department of a teaching hospital in Malaysia. BMC Health Services Research. 2020;20(1):1-7.

Comment # 6: To improve the justification of the article, I suggest that you address the importance and consequences (negative impacts) that prescription errors can cause in an emergency department.

Response: Thank you for this highly important suggestion. Unfortunately, while the negative of impact medication errors including prescription errors are well known and could be associated with negative health outcomes and economic burden, we could not find any study that evaluated the direct or indirect negative consequences associated with prescription errors in emergency departments. Therefore, we have added this as one of the research implications and its importance for future research (Page 14, lines 330-338).

*Methods

Comment # 7: The methods section is missing key information about the study.

- Did the authors collect sociodemographic information, illnesses or complaints from the patient or why he / she sought the emergency department?

Response: The demographic information collected from the patient was age, gender, weight, and height. No information was collected about the illnesses or complaints he/she sought the emergency department. This is one of the study limitations (Page 14, lines 343-346).

Comment # 8: a fundamental question: Have data been collected on the positive impact of preventing errors? Did the prevented errors save costs for the Emergency Department or for the patient? Did the prevented errors improve the patients' quality of life? Did mistakes prevent new health problems in patients' lives?

Response: Thank you for this excellent comment. No information was collected about the positive impact of preventing prescribing errors, cost saving for the Emergency Department or the patient, patients' quality of life, preventable new health problems in patients' lives as this study did not aim to explore these important points. We have added these questions as future research implications (Page 14, lines 326-330) and as one of the limitations. 

Comment # 9: Clarify who the research team is.

Response: It has been replaced by “the authors” to make it clearer to the readers.

*Results

Comment # 10: It would be very important for the authors to present the clinical, eco-economic and humanistic impacts of preventing errors. 

Response: Again we thank you for raising our awareness towards this important point which is exploring the clinical, eco-economic and humanistic impacts of preventing prescribing errors. Unfortunately, while studies published in the literature that evaluated the clinical, eco-economic and humanistic impacts of preventing prescription errors in the emergency department are limited, we also failed to assess these important domains since that the aim of this study did not included these points. Yet, we are highly encouraged to conduct research in the near future to evaluate the clinical, eco-economic and humanistic impacts of preventing prescription errors in the emergency department.

Comment # 11: Table 3: change “contributing factors” to “contributing factors (systems related)” and “causes for identified prescription errors” to “causes for identified prescription errors (human-related)”.

Response: Recommended changes by the reviewer have been made in Table 3. 

*Discussion

Comment # 12: Overall, the discussion section requires clear connection with the study findings. Further tightening the connection between results with discussion will provide clear understanding of the study contribution and future direction of prevention of medication errors in Emergency Department.

Response: Thank you for the suggestion and with the provided comments from other reviewers, the discussion now should provide a clearer read with a tighter connection between the study main findings as well as its future implications on prescribing errors and the healthcare system as well as patients related health outcomes. 

Comment # 13: It would be very important to relate the complaint or reason for going to the emergency department with the use of medicines.

Response: Thank you for this excellent comment. Unfortunately, no information was collected about the illnesses or complaints he/she sought the emergency department. This is one of the study limitations we added (Page 14, lines 329-330).

Comment # 14: If clinical, economic and humanistic impacts have not been collected, I suggest adding it as a study limitation 

Response: We have added these importantly raised points as future research implications (Page 14, lines 330-338) and as one of the limitations (Page 14, lines 326-329). 

Comment # 15: The phrase “Although this study is one of few studies that evaluated the PEs in the ED” (line 265) is inaccurate information. There are several studies in the literature.

Response: thank you for this comment and we have removed this statement. 

*Conclusion

Comment # 16: The conclusion is very short. Explore further the conclusions that you observed with the results and describe the implications that this can have.

Response: Thank you for this comment to improve the conclusion part. The conclusion was reviewed and updated accordingly. 

*References

Comment # 17: Only 5 (22,7%) studies were published in the last 5 articles. This is a problem.

Response: We have updated the reference, thanks a lot for the comment.

---

## [Decision Letter · Decision Letter 1]

9 Jul 2020

PONE-D-20-06984R1

Prescriptions Errors Prevalence and Associated Factors in an Emergency Department: A Prospective Cross-Sectional Study

PLOS ONE

Dear Dr. Alhawassi,

Thank you for submitting your manuscript to PLOS ONE. After careful consideration, we feel that it has merit but does not fully meet PLOS ONE’s publication criteria as it currently stands. Therefore, we invite you to submit a revised version of the manuscript that addresses the points raised during the review process.

Please, address all the comments raised by the reviewers. Besides, I have also several remarks with respect to the manuscript:

First, I am not sure if the unit of analysis is the patient or the prescription. It is not clear to me if the study sample comprised 504 prescriptions or 504 patients. Are the prescriptions always unique per patient or can the doctor prescribe more than one medication to one patient in each visit?I still hold my comments about the total number of medication errors. According to the text, there were 68 (again, prescriptions or patients?). However, in table 2 the total sums of therapeutic agents are only 62.Some additional data are needed to put results into context. To compare the prevalences of prescribing errors (PE) in children and adults, the number of written prescriptions for each age group must also be presented. Likewise, to compare the types of physician responsible for PEs, the total number of prescriptions made by each group is needed. Namely, in each group not only the percentages of total errors should be shown, but also the percentages of prescriptions with errors. In addition, as reviewer 3 highlights, the statistical significance of the differences must be presented.In page 9 lines 210-1, the authors describe the age and sex of patients with PE, but not in the total population. Consequently, these data provide very little information. Moreover, in my view, it will be very likely that age follows a bimodal distribution (the patients at greater risk are children and older patients) so it does not seem appropriate to present measures of central tendency. In any case, standard deviation (SD) is a measure of dispersion of data about the mean, while interquartile range (IQR) describes the spread about the median.In the 2nd paragraph of Discussion section (page 11, lines 249-54) the authors point at causes of PEs in pediatric patients. Their explanations seem very plausible, but they are not based on the results. In such section (table 1) the PEs are shown in the total population. If the authors consider that this issue is relevant, they must present the values of the different age groups in the results section.In limitations, page 14 lines 326-8, the authors should also include that the categorization of medication errors according to the severity of the outcome in the patient was not taken into account in this study.I also observe several minor points:Revise the percentages in text. In page 9, lines 213-6, the sum of percentages of PEs in aduIts is 59.9%. In page 10, lines 233-4, the sum of prescribers is 95% instead of 100.The title of table 3 includes an asterisk but not the corresponding note.Please, avoid the use of ellipses when describing the variables of the study (page 8, lines 193 and 196)

We look forward to receiving your revised manuscript.

Kind regards,

Juan F. Orueta, MD, PhD

Academic Editor

PLOS ONE

Reviewers' comments:

Reviewer's Responses to Questions

**Comments to the Author**

1. If the authors have adequately addressed your comments raised in a previous round of review and you feel that this manuscript is now acceptable for publication, you may indicate that here to bypass the “Comments to the Author” section, enter your conflict of interest statement in the “Confidential to Editor” section, and submit your "Accept" recommendation.

Reviewer #1: (No Response)

Reviewer #3: All comments have been addressed

2. Is the manuscript technically sound, and do the data support the conclusions?

Reviewer #1: Yes

Reviewer #3: Yes

3. Has the statistical analysis been performed appropriately and rigorously? 

Reviewer #1: Yes

Reviewer #3: No

4. Have the authors made all data underlying the findings in their manuscript fully available?

Reviewer #1: Yes

Reviewer #3: No

5. Is the manuscript presented in an intelligible fashion and written in standard English?

Reviewer #1: Yes

Reviewer #3: Yes

6. Review Comments to the Author

Reviewer #1: Thank you for the authors' for addressing my comments. Figure 1 has helped clarify the data collection process much clearly. I just have one very minor comment:

-page 8- line 182-192: it is a very long sentence that could be divided for clarity-

-Page 8-line 190- why only the Just culture? what about the learning culture etc? is the reporting of the error or near miss anonymised as well (for the person reporting it as well as the prescriber who nearly committed a PE)?

Few verb tense errors:

line 161: were

line 179:were

line 184: which then was recorded

line 187: submitted

Reviewer #3: Although authors have addressed most of the comments raised by the reviewers, there are still a few comments that need to be addressed:

Page 9, Line 208: "The total number of errors per patients was 82 (ratio 1:1.3) where . . .".

Total no. of errors per patient should be 82 errors/ 68 patients = 1.205.

Page 10, Lines 234-236: "88.0% of the identified PEs in this study were resolved by the pharmacist and were recorded as accepted interventions, while physicians have rejected 12.0 % of the raised recommendations."

In these 12% rejected interventions, Pharmacist had re-evaluated the prescription to categorize these into [1. Revised with no PE] [2. identified PE, not resolved].

Authors should clearly portray these figures in the Results section.

In table 1: One more row can be added in the end for total no. of prescribing errors.

Statistical significance was not shown for any data in the entire manuscript.

There are a few grammatical errors such as:

Abstract, Line 50: "About 36.8% of identified PEs was (were) related to. . ."

Page 5, Line 121: ". . . opened around (round) the corner. . ."

Page 6, Line 146: ". . .and who was (were) . . ."

Page 8, Line 179: "Identified potential PEs was (were) then discussed. . ."

Page 9, Line 208: ". . . errors per patients (patient) was . . ."

Page 14, Line 336: "PEs in the emergency setting is (are) common. . ."

7. PLOS authors have the option to publish the peer review history of their article (what does this mean?). If published, this will include your full peer review and any attached files.

Reviewer #1: **Yes: **Nada Atef Shebl

Reviewer #3: **Yes: **Mir Shoebulla Adil

---

## [Author Response · Author response to Decision Letter 1]

23 Aug 2020

We would like to thank the editor and the reviewers for the time and effort they spent on reviewing our revised manuscript entitled “Prescriptions Errors Prevalence and Associated Factors in an Emergency Department: A Prospective Cross-Sectional Study", their valuable and insightful comments have improved our manuscript substantially.

We are excited to have been given the opportunity to revise our manuscript and respond to the revisions. We have gone through all comments received and appropriate changes/amendments have been made correspondingly to the paper (Highlighted) are summarized in the following:

Editor Comments to Author

In addition to the remarks of the reviewers, I observe other points that need to be clarified 

Comment # 1: First, I am not sure if the unit of analysis is the patient or the prescription. It is not clear to me if the study sample comprised 504 prescriptions or 504 patients. Are the prescriptions always unique per patient or can the doctor prescribe more than one medication to one patient in each visit? 

Response: Thank you for this valuable comment. We have clarified this point in the manuscript “A total of 504 prescriptions were included” (Page 2, line 48; Page 8, Line 202). And yes, the prescriptions always unique per patient.

Comment # 2: I still hold my comments about the total number of medication errors. According to the text, there were 68 (again, prescriptions or patients?). However, in table 2 the total sums of therapeutic agents are only 62. 

Response: We appreciate your valuable comment. The total number of medication errors in table 2 is 62 because it does not include the 6 cases of the wrong patient in table one) (Page 8, Line 200).

Comment # 3: Some additional data are needed to put results into context. To compare the prevalence of prescribing errors (PE) in children and adults, the number of written prescriptions for each age group must also be presented. 

Likewise, to compare the types of physician responsible for PEs, the total number of prescriptions made by each group is needed. Namely, in each group not only the percentages of total errors should be shown, but also the percentages of prescriptions with errors.

 In addition, as reviewer 3 highlights, the statistical significance of the differences must be presented.

Response: Thank you for pointing out this critical point. We agree with the reviewer; unfortunately, we have not collected information about the total number of written prescriptions for each age group. We have collected the demographic information only for prescriptions with errors; therefore, we could not calculate the prevalence. Besides, we have not collected the data of the type of prescriber for prescriptions without errors; therefore, we have added this point as one of the study limitations (Page 13, lines 328-330). In regards to the number of prescriptions errors made by each type of physician, we have added the numbers and the percentage in Table 4. 

The reason for not adding the statistical significance for the study data was that the sample size was small, and many cells have zero counts when we do the chi-square test to measure the statistical significance. Therefore, a larger sample size is needed to provide inferential statistics.

Comment # 4: 

In page 9 lines 210-1, the authors describe the age and sex of patients with PE, but not in the total population. Consequently, these data provide very little information. 

Moreover, in my view, it will be very likely that age follows a bimodal distribution (the patients at greater risk are children and older patients) so it does not seem appropriate to present measures of central tendency. In any case, standard deviation (SD) is a measure of dispersion of data about the mean, while interquartile range (IQR) describes the spread about the median. 

Response: Thanks for this valuable point. We agree with the reviewer; unfortunately, we have not collected information about the age or sex for prescriptions without PEs. We have added this point as one of the study limitations (Page 13, lines 328-330).

As per the reviewer suggestion, we have removed the measure of central tendency and used the percentages and number to present the categorical data.

Comment # 5: In the 2nd paragraph of Discussion section (page 11, lines 249-54) the authors point at causes of PEs in pediatric patients. Their explanations seem very plausible, but they are not based on the results. In such section (table 1) the PEs are shown in the total population. If the authors consider that this issue is relevant, they must present the values of the different age groups in the results section. ***

Response: Thank you for the comment. Now we added to the results section the following sentence “It has to be noted that wrong strength/concentration, wrong dosage form, and wrong route of administration was higher in the adults’ population as compared to the pediatrics.” (Page 9, lines 209-210)

Comment 6: In limitations, page 14 lines 326-8, the authors should also include that the categorization of medication errors according to the severity of the outcome in the patient was not taken into account in this study.

Response: Thank you for the suggestion; we have added this point as one of the limitations (Page 13, lines 326-328).

Comment # 7: I also observe several minor points. Revise the percentages in text. In page 9, lines 213-6, the sum of percentages of PEs in aduIts is 59.9%. In page 10, lines 233-4, the sum of prescribers is 95% instead of 100.

Response: Thank you for this comment. We have now added to the text as well as the table the percentage of all prescribers (Page 9, lines 227-229).

Table 4: Personal involved in prescription errors 

Job Title %

Resident 39.4

Specialist 30.3

General Practitioner 24.0

Nurse 4.5

Others 1.5

Comment # 8: The title of table 3 includes an asterisk but not the corresponding note.

Response: We have removed the asterisk as no corresponding note is needed.

Comment # 79: Please, avoid the use of ellipses when describing the variables of the study (page 8, lines 193 and 196)

Response: Thank you for noticing this typo, we have now corrected it appropriately (Page 8, 189 and 192).

Reviewer 1 Comments to Author

Thank you for the authors' for addressing my comments. Figure 1 has helped clarify the data collection process much clearly. I just have one very minor comment

Comment 1: Page 8- line 182-192: it is a very long sentence that could be divided for clarity

Response: Thank you for the suggestion, now we have divided the sentence to make to improve the clarity to the readers (Page 7 and 8, lines 180-188).

Comment 2: Page 8-line 190- why only the Just culture? what about the learning culture etc? is the reporting of the error or near miss anonymised as well (for the person reporting it as well as the prescriber who nearly committed a PE)?

Response: Thank you for this comment, the reporting of the error or near miss were not anonymized, therefore we have changed it to learning culture (Page 8, line 187). 

Comment 3: Few verb tense errors:

line 161: were

line 179:were

line 184: which then was recorded

line 187: submitted

Response: Thank you for noticing these grammatical mistakes, all corrections have been made.

Reviewer 3 Comments to Author

Although authors have addressed most of the comments raised by the reviewers, there are still a few comments that need to be addressed:

Comment 1: Page 9, Line 208: "The total number of errors per patients was 82 (ratio 1:1.3) where . . .". Total no. of errors per patient should be 82 errors/ 68 patients = 1.205.

Response: Thank you for the correction (Page 8, line 204).

Comment 2: Page 10, Lines 234-236: "88.0% of the identified PEs in this study were resolved by the pharmacist and were recorded as accepted interventions, while physicians have rejected 12.0 % of the raised recommendations."

In these 12% rejected interventions, Pharmacist had re-evaluated the prescription to categorize these into [1. Revised with no PE] [2. identified PE, not resolved].

Authors should clearly portray these figures in the Results section.

Response: Thank you for pointing this out. As per the reviewer recommendation, we have added this sentence to the results section (Page 10, lines 232-233).

Comment 3: In table 1: One more row can be added in the end for total no. of prescribing errors.

Response: Thank you for the suggestion. Now we have added the total no. of prescribing errors in the first raw of Table 1.

Comment 4: Statistical significance was not shown for any data in the entire manuscript.

Response: Thank you for pointing this out. The reason for not adding the statistical significance for the study data was that the sample size was small, and many cells have zero counts when we do the chi-square test to measure the statistical significance. Therefore, a larger sample size is needed to provide inferential statistics.

Comment 5: There are a few grammatical errors such as:

Abstract, Line 50: "About 36.8% of identified PEs was (were) related to. . ."

Page 5, Line 121: ". . . opened around (round) the corner. . ."

Page 6, Line 146: ". . .and who was (were) . . ."

Page 8, Line 179: "Identified potential PEs was (were) then discussed. . ."

Page 9, Line 208: ". . . errors per patients (patient) was . . ."

Page 14, Line 336: "PEs in the emergency setting is (are) common. . ."

Response: Thank you for noticing these grammatical mistakes, all corrections have been made.

---

## [Decision Letter · Decision Letter 2]

14 Oct 2020

PONE-D-20-06984R2

Prescriptions Errors Prevalence and Associated Factors in an Emergency Department: A Prospective Cross-Sectional Study

PLOS ONE

Dear Dr. Alhawassi,

Thank you for submitting your manuscript to PLOS ONE. After careful consideration, we feel that it has merit but does not fully meet PLOS ONE’s publication criteria as it currently stands. Therefore, we invite you to submit a revised version of the manuscript that addresses the points raised during the review process.

First of all, I must explain that there has been a change of reviewers. The new referees have submitted their evaluations and valuable comments. Although most of their observations agree with the previous ones, you will perceive some differences.

PLOS ONE evaluates submitted works based on methodological rigor, regardless of perceived novelty. However, the manuscripts must include references to other similar previous studies. Please, take into account the recommendations of the reviewers and quote those papers.

The authors have been responsive to most of the previous comments and improved the manuscript. Though, there are still some points that should be amended to clarify the description of methods and presentation of results. Besides the comments of the reviewers, I would suggest (as a minor point) to show the same number of decimal places in the percentages in the tables (for example, only one).

We look forward to receiving your revised manuscript.

Kind regards,

Juan F. Orueta, MD, PhD

Academic Editor

PLOS ONE

Reviewers' comments:

Reviewer's Responses to Questions

**Comments to the Author**

1. If the authors have adequately addressed your comments raised in a previous round of review and you feel that this manuscript is now acceptable for publication, you may indicate that here to bypass the “Comments to the Author” section, enter your conflict of interest statement in the “Confidential to Editor” section, and submit your "Accept" recommendation.

Reviewer #4: (No Response)

Reviewer #5: (No Response)

2. Is the manuscript technically sound, and do the data support the conclusions?

Reviewer #4: Partly

Reviewer #5: Yes

3. Has the statistical analysis been performed appropriately and rigorously? 

Reviewer #4: I Don't Know

Reviewer #5: Yes

4. Have the authors made all data underlying the findings in their manuscript fully available?

Reviewer #4: No

Reviewer #5: No

5. Is the manuscript presented in an intelligible fashion and written in standard English?

Reviewer #4: Yes

Reviewer #5: Yes

6. Review Comments to the Author

Reviewer #4: The study highlights key health issue in the region.

For a start, I suggest the authors to quote and describe previous similar studies (done in the region) in the introduction section. There are two or three studies in Iran, one in Oman etc. There is a more recent systematic review about MEs in Middle East (2019), the one referenced in text is published in 2013.

Link: https://link.springer.com/article/10.1007/s00228-019-02689-y

The current status of results do not provide enough insight into the problem of PEs in Saudi Arabia. One way is to revisit data analysis, which can be improved by investigating the predictors for PEs. For example, class of prescribers, length of stay in ED, number of medications in prescription order.. etc.

Line 103: Ref. 13: There is more recent review about MEs (as mentioned above).

Line 196: SPSS should be referenced in text as (IBM Corp., Armonk, N.Y., USA).

Table 1: number and % does not correlate with each other. For PE type (wrong dose): 19/82 = 23% but in table is 20.7%!!

Table 2: It might be good to group all oral formulation as one (capsule, tablet etc..)

Table 4: Title should be changed to (Healthcare professionals involved in PEs)

Appendix: It would be useful to look at the data collection tool used for this study, to be included as appendix.

Reviewer #5: The authors have mostly addressed the reviewers' comments satisfactorily. However, there a couple of issues remain:

- One of the reviewers raised the issues of the denominator. This is still confusing in the manuscript. From what I can see, there were 504 prescriptions assessed, there were 82 errors in 68 prescriptions. But the number of patients for whom the 504 prescriptions were written is not presented. I suggest this be inserted in the first sentence of the Results e.g. A total of 504 prescriptions for xx patients were assessed.

- Results, second sentence: "The total number of errors per patient was 82 (ratio 1:205)" . Firstly, it's impossible there were 82 errors per patient. There was a total of 82 errors. How many patients were there? Furthermore, "1:205" cannot be correct. I think the reviewer's suggestion was 1.205 - i.e. 1.205 errors per patient (not 82 errors per patient) - is this correct?

- Table 1 states the total number of PEs is 82, but the addition of the types of errors is 77. If this table is counting PEs, the total should be 82. Please check.

- Table 2 - one reviewer raised the issue of the numbers not adding up to the total number of errors. The explanation given by the authors for this discrepancy is satisfactory, but needs clarification within the table by a footnote as readers will ask the same question. Same for Table 3 where the system related factors don't add up to 82.

- When presenting the results as a prevalence it is ambiguous what the denominator is. I suggest revising this to be e.g. 13.5% of prescription had at least one error, rather than saying a prevalence of 13.5% (as in Discussion first paragraph). This also applies when discussing other studies e.g. Discussion, paragraph 1 - prevalence of 13.4%... is this 13.4 errors per 100 orders or 100 patients, or 13.4% of prescriptions with at least one error?

-Discussion, line 326-329: The main limitation that should be listed here is the lack of a denominator for the data provided. So for example, the most frequent medications with errors were analgesics and antibiotics, probably because these were the most commonly prescribed drugs. A comparison of error rates would provide a more complete picture of what types of prescriptions were more likely to have errors. However, as the authors have acknowledged that denominators were not available, this should be mentioned as a limitation more generally with the data and not only with the lack of denominators for the age and gender distribution (as it is currently worded).

-From the Methods, I understand that discharge prescriptions were assessed for errors and not inpatient medication charts. This is based on the Study sample section of the Methods. If so, I think this should be made explicit in the Abstract and even the title.

Some more minor issues:

- Table 1 has an asterisk not addressed with a footnote. Please provide full form of abbreviation PE in the table footnote.

- Table 3 has an asterisk not addressed with a footnote.

-Table 4 needs the number of errors presented in addition to percentages (as in the other tables)

7. PLOS authors have the option to publish the peer review history of their article (what does this mean?). If published, this will include your full peer review and any attached files.

Reviewer #4: No

Reviewer #5: No

---

## [Author Response · Author response to Decision Letter 2]

28 Oct 2020

We want to thank the editor and the reviewers for the time and effort they spent on reviewing our revised manuscript entitled “Prescriptions Errors Prevalence and Associated Factors in an Emergency Department: A Prospective Cross-Sectional Study", their valuable and insightful comments have improved our manuscript substantially.

Reviewer #4 Comments to Author

The study highlights key health issue in the region

Comment 1: For a start, I suggest the authors to quote and describe previous similar studies (done in the region) in the introduction section. There are two or three studies in Iran, one in Oman etc. There is a more recent systematic review about MEs in Middle East (2019), the one referenced in text is published in 2013. Link: https://link.springer.com/article/10.1007/s00228-019-02689-y

Response: Thank you for suggesting adding to the introduction similar studies and a recent systematic review. Now, we have added these references to the introduction (Page 4, lines 99-101, 107-110).

References:

1. Izadpanah F, Nikfar S, Imcheh FB, Amini M, Zargaran M. Assessment of frequency and causes of medication errors in pediatrics and emergency wards of teaching hospitals affiliated to Tehran University of Medical Sciences (24 Hospitals). Journal of Medicine and Life. 2018;11(4):299.

2. Thomas B, Paudyal V, MacLure K, Pallivalapila A, McLay J, El Kassem W, et al. Medication errors in hospitals in the Middle East: a systematic review of prevalence, nature, severity and contributory factors. European journal of clinical pharmacology. 2019;75(9):1269-82.

Comment 2: The current status of results do not provide enough insight into the problem of PEs in Saudi Arabia. One way is to revisit data analysis, which can be improved by investigating the predictors for PEs. For example, class of prescribers, length of stay in ED, number of medications in prescription order.. etc.

Response: Thank you for this comment. We totally agree with the reviewer; however, this information is not available for the researchers and we are limited by the data we have in this study.

Comment 3: Line 103: Ref. 13: There is more recent review about MEs (as mentioned above).

Response: Thank you for pointing this out, now we have added the recent systematic review to the introduction (Page 4, lines 107-110).

Comment 4: Line 196: SPSS should be referenced in text as (IBM Corp., Armonk, N.Y., USA).

Response: Now we have referenced SPSS in the text, thanks (Page 8, line 200).

Comment 5: Table 1: number and % does not correlate with each other. For PE type (wrong dose): 19/82 = 23% but in table is 20.7%!!

Response: Thank you for this comment as it made us revisit the original data and found missing categories. Now the table is updated as well as the % in the abstract (Page 19, Table 1; Page 2, lines 50-51).

Comment 6: Table 2: It might be good to group all oral formulation as one (capsule, tablet etc..)

Response: Thank you for the suggestion; we have made the suggestion to Table 2.

Comment 7: Table 4: Title should be changed to (Healthcare professionals involved in PEs)

Response: Thank you for the suggestion. Now we have made the suggested changes in Table 4.

Comment 8: Appendix: It would be useful to look at the data collection tool used for this study, to be included as appendix.

Response: Thank you for the suggestion. Now we have added the data collection tool (Appendix I) (Page 22).

Reviewer #5 Comments to Author

The authors have mostly addressed the reviewers' comments satisfactorily. However, there a couple of issues remain:

Comment 1: One of the reviewers raised the issues of the denominator. This is still confusing in the manuscript. From what I can see, there were 504 prescriptions assessed, there were 82 errors in 68 prescriptions. But the number of patients for whom the 504 prescriptions were written is not presented. I suggest this be inserted in the first sentence of the Results e.g. A total of 504 prescriptions for xx patients were assessed.

Response: Thank you for pointing this out. Please note that the total number of patients (Prescriptions) included in the study is 504 out of this number 68 Patients (Prescription) was identified to have errors; therefore, each patient had one prescription. Now we have added this to the results section as well as the abstract (Page 9, lines 204-205).

Comment 2: Results, second sentence: "The total number of errors per patient was 82 (ratio 1:205)" . Firstly, it's impossible there were 82 errors per patient. There was a total of 82 errors. How many patients were there? Furthermore, "1:205" cannot be correct. I think the reviewer's suggestion was 1.205 - i.e. 1.205 errors per patient (not 82 errors per patient) - is this correct?

Response: Thank you for noticing this, we have rephrased the sentence now (Page 10, Line 243)

Comment 3: Table 1 states the total number of PEs is 82, but the addition of the types of errors is 77. If this table is counting PEs, the total should be 82. Please check.

- Table 2 - one reviewer raised the issue of the numbers not adding up to the total number of errors. The explanation given by the authors for this discrepancy is satisfactory, but needs clarification within the table by a footnote as readers will ask the same question. Same for Table 3 where the system related factors don't add up to 82.

Response: Now the table is updated as well as the % in the abstract (Page 19, Table 1; Page 2, lines 50-51). 

Regarding Table 3, we have added a footnote to clear the confusion (Page 21, Table-3).

Comment 4: When presenting the results as a prevalence it is ambiguous what the denominator is. I suggest revising this to be e.g. 13.5% of prescription had at least one error, rather than saying a prevalence of 13.5% (as in Discussion first paragraph). This also applies when discussing other studies e.g. Discussion, paragraph 1 - prevalence of 13.4%... is this 13.4 errors per 100 orders or 100 patients, or 13.4% of prescriptions with at least one error?

Response: Thank you for the correction. We totally agree with the reviewer and since we don’t have a denominator to calculate the prevalence, now we have removed the term “Prevalence” and replaced it in the discussion part (Page 10, line 243).

Comment 5: Discussion, line 326-329: The main limitation that should be listed here is the lack of a denominator for the data provided. So for example, the most frequent medications with errors were analgesics and antibiotics, probably because these were the most commonly prescribed drugs. A comparison of error rates would provide a more complete picture of what types of prescriptions were more likely to have errors. However, as the authors have acknowledged that denominators were not available, this should be mentioned as a limitation more generally with the data and not only with the lack of denominators for the age and gender distribution (as it is currently worded).

Response: As per the reviewer’s suggestion, we have added this limitation to the discussion part (Page 14, lines 334-336).

Comment 6: From the Methods, I understand that discharge prescriptions were assessed for errors and not inpatient medication charts. This is based on the Study sample section of the Methods. If so, I think this should be made explicit in the Abstract and even the title.

Response: Thank you for pointing this out, we have now added “discharge prescriptions” to the title as well as the abstract.

Comment 7: Some more minor issues:

- Table 1 has an asterisk not addressed with a footnote. Please provide full form of abbreviation PE in the table footnote.

- Table 3 has an asterisk not addressed with a footnote.

-Table 4 needs the number of errors presented in addition to percentages (as in the other tables)

Response: Thank you for pointing these out. Recommended changes have been made.

---

## [Decision Letter · Decision Letter 3]

16 Nov 2020

PONE-D-20-06984R3

Prescribing Errors and Associated Factors in Discharge Prescriptions in the Emergency Department: A Prospective Cross-Sectional Study

PLOS ONE

Dear Dr. Alhawassi,

Thank you for submitting your manuscript to PLOS ONE. After careful consideration, we feel that it has merit but does not fully meet PLOS ONE’s publication criteria as it currently stands. The authors have been responsive to most previous comments, but there are still minor but relevant remarks.Therefore, we invite you to submit a revised version of the manuscript that addresses the points raised during the review process.

The reviewer points out that the manuscript requires careful proofreading. Besides her comments, revise the whole text. In several sentences, the wording blurs the meaning. For example:

- Page 9, lines 203-9. It seems clear that 68 prescriptions contained one or more errors. As 11 (16%) of them contained more than one error, the total number of identified PEs was 82. However, the wording is not clear, and specifically the expression “total number of errors per patient” is confusing.

- According to the study, 36.8% of the identified PEs occurred in pediatric patients. However, it does not mean that “36.8% of prescriptions in the pediatric setting had at least one PEs”, as the authors assert. (Page 11, line 250)

Also, other parts require changes:

- In the discussion, there are two paragraphs (page 12, lines 276-81; page 13 line, 304 and following) that could be merged and shortened to avoid redundancy. Both of them describe studies reporting that the presence of pharmacists in ED results in a reduction of PEs. Also, a citation is required to support the first sentence of the paragraph on page 12.

- The limitations of the study are explained in the corresponding section. However, this paragraph should be reorganized. A call for further research is frequently included at the end of the discussion section, but not in the middle of the limitations (page 14, line 329-30). So, it is clear that in previous drafts this point was the end of the section and, later, other sentences were added. Also, it is not a felicitous remark to conclude that “Consequently, these data provide very little information” (line 335). Please, rewrite the whole paragraph.

We look forward to receiving your revised manuscript.

Kind regards,

Juan F. Orueta, MD, PhD

Academic Editor

PLOS ONE

Reviewers' comments:

Reviewer's Responses to Questions

**Comments to the Author**

1. If the authors have adequately addressed your comments raised in a previous round of review and you feel that this manuscript is now acceptable for publication, you may indicate that here to bypass the “Comments to the Author” section, enter your conflict of interest statement in the “Confidential to Editor” section, and submit your "Accept" recommendation.

Reviewer #5: (No Response)

2. Is the manuscript technically sound, and do the data support the conclusions?

Reviewer #5: Yes

3. Has the statistical analysis been performed appropriately and rigorously? 

Reviewer #5: Yes

4. Have the authors made all data underlying the findings in their manuscript fully available?

Reviewer #5: No

5. Is the manuscript presented in an intelligible fashion and written in standard English?

Reviewer #5: No

6. Review Comments to the Author

Reviewer #5: The authors have addressed most of the reviewer comments. One remains to be addressed:

-Table 2 - one reviewer raised the issue of the numbers not adding up to the total number of errors. The explanation given by the authors for this discrepancy is satisfactory, but needs clarification within the table by a footnote as readers will ask the same question.

-One other minor comment is to explain what is meant by Drops in Table 2. e.g. is this eye drops, nasal drops??

Some grammar changes needed (but please check rest of manuscript carefully):

-Abstract, Methods, first sentence: "in an ambulatory ED", not "at ambulatory ED".

-Abstract, Methods, second sentence: "Data were collected for six month using a customized reporting tool."; not "Data collected for six months period...".

-Abstract, Methods, Third sentence: "All patients discharged from the ED with a discharge prescription..."; not "All patients who were discharged from ED with a discharged prescription..."

-Abstract, Results, first sentence: "504 prescriptions for 504 patients were reviewed and 13.5% (n=68) had at least one error" or "13.5% (n=68) of the 504 prescriptions reviewed (for 504 patients) had at least one error".

-Abstract, Results, second sentence: move the % to after the error type e.g. "Main PE encountered were wrong dose (23.2%), wrong frequency (20.7%)....".

-Abstract, Conclusion: "PEs in the ED are common...", not "PEs in ED setting is common..."

-Introduction, page 4, lines 99-101 (new text): "... revealed that 50.5% of the total MEs occurred in the ED, with 22.6% of these being PEs."

-Introduction, page 4, lines 107-109 (new text): the sentence beginning "A recent review..." is not a full sentence, pls revise.

-Discussion, first sentence: "This study found that 13.5% of patients' electronic prescriptions in the ED had at least one PE." End the sentence here then compare to the literature in a new sentence, as this is currently a very long sentence.

-Discussion, first sentence: "... where PEs were found to occur in 13.4% to 50.5% of prescriptions."

-Discussion, first paragraph, last sentence: "...by several studies, but PEs are nonetheless alarmingly common in the ED".

-Discussion, last paragraph, line 333-335: "For example, a comparison of error rates for each medication class would...".

7. PLOS authors have the option to publish the peer review history of their article (what does this mean?). If published, this will include your full peer review and any attached files.

Reviewer #5: No

---

## [Author Response · Author response to Decision Letter 3]

8 Dec 2020

We want to thank the editor and the reviewer for the time and effort they spent reviewing our revised manuscript entitled “Prescribing Errors and Associated Factors in Discharge Prescriptions in the Emergency Department: A Prospective Cross-Sectional Study", their valuable and insightful comments have improved our manuscript substantially.

Editor Comments to Author

The reviewer points out that the manuscript requires careful proofreading. Besides her comments, revise the whole text. In several sentences, the wording blurs the meaning. 

Comment 1: Page 9, lines 203-9. It seems clear that 68 prescriptions contained one or more errors. As 11 (16%) of them contained more than one error, the total number of identified PEs was 82. However, the wording is not clear, and specifically the expression “total number of errors per patient” is confusing.

Response: Thank you for pointing this out; now we have improved the clarity of this sentence. 

Comment 2: According to the study, 36.8% of the identified PEs occurred in pediatric patients. However, it does not mean that “36.8% of prescriptions in the pediatric setting had at least one PEs”, as the authors assert. (Page 11, line 250)

Response: Thank you for this comment. We have removed “at least one”, as we meant the identified PEs.

Comment 3: In the discussion, there are two paragraphs (page 12, lines 276-81; page 13 line, 304 and following) that could be merged and shortened to avoid redundancy. Both of them describe studies reporting that the presence of pharmacists in ED results in a reduction of PEs. Also, a citation is required to support the first sentence of the paragraph on page 12.

Response: Thank you for pointing this out; we have now merged and shortened these paragraphs.

Comment 4: The limitations of the study are explained in the corresponding section. However, this paragraph should be reorganized. A call for further research is frequently included at the end of the discussion section, but not in the middle of the limitations (page 14, line 329-30). So, it is clear that in previous drafts this point was the end of the section and, later, other sentences were added. Also, it is not a felicitous remark to conclude that “Consequently, these data provide very little information” (line 335). Please, rewrite the whole paragraph.

Response: We agree with the editor; we have removed the call for future research from the limitations part and include it at the end of the discussion part. 

Reviewer #5 Comments to Author

The authors have addressed most of the reviewer comments. One remains to be addressed:

Comment 1: Table 2 - one reviewer raised the issue of the numbers not adding up to the total number of errors. The explanation given by the authors for this discrepancy is satisfactory, but needs clarification within the table by a footnote as readers will ask the same question.

Response: Thank you for pointing this out. We have included a footnote explaining this point. 

Comment 2: One other minor comment is to explain what is meant by Drops in Table 2. e.g. is this eye drops, nasal drops??

Response: Thank you for noticing this, we have changed it into “Eye Drops”.

Comment 3: Some grammar changes needed (but please check rest of manuscript carefully):

-Abstract, Methods, first sentence: "in an ambulatory ED", not "at ambulatory ED".

-Abstract, Methods, second sentence: "Data were collected for six month using a customized reporting tool."; not "Data collected for six months period...".

-Abstract, Methods, Third sentence: "All patients discharged from the ED with a discharge prescription..."; not "All patients who were discharged from ED with a discharged prescription..."

-Abstract, Results, first sentence: "504 prescriptions for 504 patients were reviewed and 13.5% (n=68) had at least one error" or "13.5% (n=68) of the 504 prescriptions reviewed (for 504 patients) had at least one error".

-Abstract, Results, second sentence: move the % to after the error type e.g. "Main PE encountered were wrong dose (23.2%), wrong frequency (20.7%)....".

-Abstract, Conclusion: "PEs in the ED are common...", not "PEs in ED setting is common..."

-Introduction, page 4, lines 99-101 (new text): "... revealed that 50.5% of the total MEs occurred in the ED, with 22.6% of these being PEs."

-Introduction, page 4, lines 107-109 (new text): the sentence beginning "A recent review..." is not a full sentence, pls revise.

-Discussion, first sentence: "This study found that 13.5% of patients' electronic prescriptions in the ED had at least one PE." End the sentence here then compare to the literature in a new sentence, as this is currently a very long sentence.

-Discussion, first sentence: "... where PEs were found to occur in 13.4% to 50.5% of prescriptions."

-Discussion, first paragraph, last sentence: "...by several studies, but PEs are nonetheless alarmingly common in the ED".

-Discussion, last paragraph, line 333-335: "For example, a comparison of error rates for each medication class would...".

Response: Thank you for the comments on the grammatical changes. All changes have been made and highlighted.

---

## [Decision Letter · Decision Letter 4]

29 Dec 2020

Prescribing Errors and Associated Factors in Discharge Prescriptions in the Emergency Department: A Prospective Cross-Sectional Study

PONE-D-20-06984R4

Dear Dr. Alhawassi,

We’re pleased to inform you that your manuscript has been judged scientifically suitable for publication and will be formally accepted for publication once it meets all outstanding technical requirements.

Kind regards,

Juan F. Orueta, MD, PhD

Academic Editor

PLOS ONE

Additional Editor Comments (optional):

Please, amend the typos in the manuscript:

Lines 256-7: Repeated sentence fragment ("This was also found high by other studies, for instance, among the ped").Lines 262-4: Meaningless sentence. Please, put the right parentheses mark in the proper placeLine 302: the term "evaluated" should be replaced by "evaluating".

Reviewers' comments:

Reviewer's Responses to Questions

**Comments to the Author**

1. If the authors have adequately addressed your comments raised in a previous round of review and you feel that this manuscript is now acceptable for publication, you may indicate that here to bypass the “Comments to the Author” section, enter your conflict of interest statement in the “Confidential to Editor” section, and submit your "Accept" recommendation.

Reviewer #5: All comments have been addressed

2. Is the manuscript technically sound, and do the data support the conclusions?

Reviewer #5: Yes

3. Has the statistical analysis been performed appropriately and rigorously? 

Reviewer #5: Yes

4. Have the authors made all data underlying the findings in their manuscript fully available?

Reviewer #5: No

5. Is the manuscript presented in an intelligible fashion and written in standard English?

Reviewer #5: Yes

6. Review Comments to the Author

Reviewer #5: The authors have addressed the reviewers' comments satisfactorily. I congratulate the authors on working through several rounds of revision patiently.

7. PLOS authors have the option to publish the peer review history of their article (what does this mean?). If published, this will include your full peer review and any attached files.

Reviewer #5: No

---

## [Editor Report · Acceptance letter]

4 Jan 2021

PONE-D-20-06984R4 

Prescribing Errors and Associated Factors in Discharge Prescriptions in the Emergency Department: A Prospective Cross-Sectional Study 

Dear Dr. Alhawassi:

I'm pleased to inform you that your manuscript has been deemed suitable for publication in PLOS ONE. Congratulations! Your manuscript is now with our production department. 

Kind regards, 

on behalf of

Dr. Juan F. Orueta 

Academic Editor

PLOS ONE